# Investigation of Bacterial Species and Their Antimicrobial Drug Resistance Profile in Feline Urinary Tract Infection in Thailand

**DOI:** 10.3390/ani15152235

**Published:** 2025-07-30

**Authors:** Kankanit Lapcharoen, Chunyaput Bumrungpun, Wiyada Chumpol, Kamonwan Lunha, Suganya Yongkiettrakul, Porntippa Lekcharoensuk, Chantima Pruksakorn

**Affiliations:** 1Department of Microbiology and Immunology, Faculty of Veterinary Medicine, Kasetsart University, Bangkok 10900, Thailand; kankanit.la@ku.th (K.L.); fvetptn@ku.ac.th (P.L.); 2Veterinary Diagnostic Laboratory, Faculty of Veterinary Medicine, Kasetsart University, Bangkok 10900, Thailand; fvetcpb@ku.ac.th; 3National Center for Genetic Engineering and Biotechnology, National Science and Technology Development Agency, Pathum Thani 12120, Thailand; wiyada.chu@ncr.nstda.or.th (W.C.); kamonwan.lun@ncr.nstda.or.th (K.L.); suganya.yon@biotec.or.th (S.Y.)

**Keywords:** antimicrobial resistance, bacteria, cat, urinary tract infections

## Abstract

This study in Bangkok, Thailand, investigated bacterial prevalence and antimicrobial resistance in feline urinary tract infections from cystocentesis samples. *Escherichia coli* was the most frequent isolate. Resistance to amoxicillin/clavulanic acid and sulfamethoxazole/trimethoprim was widespread among key pathogens. High rates of methicillin resistance in *Staphylococcus* species, and multidrug resistance across Gram-negative, *Staphylococcus*, and *Enterococcus* species, severely complicate empirical treatment and raise considerable concerns about antimicrobial resistance in cats.

## 1. Introduction

Bacterial urinary tract infections (UTIs) are common in companion animals, but occur less frequently in cats than in dogs [1,2]. However, European epidemiological studies report a high incidence of bacterial UTIs in cats, with prevalence rates ranging from 8 to 25% [3,4]. Variations in geographic, climate, and other factors may affect the incidence of UTIs in cats. Bacteria causing cystitis often originate in the distal urethra and ascend into the normally sterile urinary bladder, potentially progressing to the upper urinary tract [5]. This process leads to inflammation and clinical signs such as pollakiuria, stranguria, dysuria, urinary incontinence, and hematuria [6,7]. Untreated or inadequately treated lower UTIs can progress to ascending infection, upper urinary tract infection, pyelonephritis, and even bacteremia [8].

Bacterial causes of feline UTIs exhibit notable variations, even within the same region over time. Studies across several European countries (e.g., Belgium, France, Sweden, Italy, and the United Kingdom) between 2013–2014 and 2017–2018 revealed *Escherichia coli* as the predominant pathogen (61.2% and 48.3% of cases, respectively). The second most common pathogen shifted from coagulase-negative staphylococci (CoNS) at 14.5% in 2013–2014 to *Enterococcus* spp. at 17.9% in 2017–2018. *Enterococcus* spp. was previously third (13.7% in 2013–2014), while CoNS fell to fourth (13.2% in 2017–2018). Coagulase-positive staphylococci (CoPS) were the fourth most common in 2013–2014 (7.6%), while *Proteus* spp. were reported as a notable pathogen at 5.3% in 2017–2018 [9]. In further European data from 2018 to 2019 in the United Kingdom, *Escherichia coli* remained the most prevalent pathogen (43.7%), followed by other Enterobacteriaceae (26.4%), *Enterococcus* spp. (14.9%), and *Staphylococcus* spp. (9.2%) [10]. In contrast, studies from Thailand highlight regional differences: Kasetsart University in 2018 reported *Staphylococcus* spp. (28%), *Escherichia coli* (22%), and *Proteus* spp. (11%) as the most common [11]. A separate study at Chiang Mai University between 2012 and 2016 identified *Pseudomonas* spp. (25%), *Escherichia coli* (20.8%), and *Proteus* spp. (16.7%) as the most frequent isolates [12]. These findings underscore that the bacterial species responsible for feline UTIs can vary significantly by region and year. Given the limited recent research on UTI-causing bacteria in cats across different parts of Thailand, collecting more comprehensive and region-specific data is crucial for developing effective and targeted treatment strategies.

Widespread antimicrobial resistance (AMR) is an increasing concern in human and veterinary medicine, particularly in small animal practice. UTIs, a common diagnosis in cats worldwide, are increasingly complicated by multidrug-resistant (MDR) bacteria [13]. MDR—defined as resistance to three or more antimicrobial classes—significantly complicates treatment and can prolong illness [14]. This growing resistance is exemplified by the emergence of AMR in bacterial isolates such as methicillin-resistant *Staphylococcus aureus* (MRSA) and MDR strains of *Escherichia coli* and *Klebsiella pneumoniae*, which pose serious concerns for both veterinary and public health [15]. These pathogens often harbor clinically and epidemiologically significant resistance mechanisms, including penicillin-binding protein 2a (PBP2a) and extended-spectrum beta-lactamases (ESBLs), that enhance their ability to evade commonly used antimicrobials. As a result, effective infection management becomes increasingly challenging in both animals and humans [13]. Specifically, ESBLs—enzymes produced by Gram-negative bacteria, most commonly *Escherichia coli* and *Klebsiella pneumoniae*—confer resistance to beta-lactam antibiotics, including newer-generation cephalosporins and monobactams [16].

The emergence of methicillin-resistant *Staphylococcus* species (MRS) further complicates treatment options. The widespread use of β-lactam antimicrobials to treat *Staphylococcus* infections has led to the emergence of MRS [17]. The primary resistance mechanism is due to the presence of the *mec*A gene, which is carried by the staphylococcal chromosomal cassette *mec* (SCC*mec*) [18]. SCC*mec* is a mobile genetic element that facilitates the spread of methicillin resistance by integrating into various genomic locations. It encodes PBP2a, a modified protein that exhibits reduced affinity for β-lactam antimicrobials, thus hindering their effectiveness in inhibiting bacterial cell wall synthesis. MRS are typically resistant to most β-lactam drugs and often resistant to multiple classes of antimicrobials [19]. Methicillin resistance is found in several *Staphylococcus* species, including methicillin-resistant *Staphylococcus aureus* (MRSA) and *Staphylococcus pseudintermedius* (MRSP). However, reports of MRS in feline UTIs remain limited. This resistance significantly reduces treatment options, often resulting in prolonged therapy, increased healthcare costs, and a greater risk of treatment failure [14,20].

Understanding the specific etiological bacterial agents and their antimicrobial susceptibility is essential for effective treatment and prevention of feline UTIs, especially amid the rise of AMR. Although empirical therapy is commonly used, its effectiveness is declining due to increasing resistance, underscoring the importance of susceptibility testing before treatment. Effective treatment of UTIs in cats relies on the selection of appropriate antimicrobials. While the International Society for Companion Animal Infectious Diseases (ISCAID) guidelines offer a valuable framework, significant regional variations exist in bacterial isolates and their antimicrobial susceptibility [2,8]. Therefore, empirical antimicrobial therapy must be guided by current local surveillance data to ensure effective drug selection against predominant regional pathogens. Such local data on bacterial prevalence and resistance, like that provided by this study, are essential for guiding drug selection, minimizing resistance, improving clinical outcomes, and supporting sustainable antimicrobial stewardship. This study, therefore, aimed to (1) determine the current prevalence of bacterial species causing UTIs in cats in Thailand; and (2) assess their AMR profiles.

## 2. Materials and Methods

### 2.1. Ethical Approval

This retrospective study focused on bacterial species and their antimicrobial susceptibility patterns, as determined from leftover diagnostic samples collected as part of standard veterinary care. No additional procedures or interventions were performed on the cats or their owners, and no personally identifying information was included in the analysis or dissemination of results. The Kasetsart University Institutional Animal Care and Use Committee (IACUC) approved this study under protocol number U1-00453-2558 on 18 September 2024.

### 2.2. Feline Urine Samples and Data Collection

Between June 2022 and May 2023, urine samples were collected via cystocentesis from cats presented to the Veterinary Teaching Hospital, Kasetsart University, Bangkok, based on the attending veterinarian’s clinical judgment. These cats exhibited lower urinary tract infection (UTI) symptoms, including one or more of the following: pollakiuria, stranguria, dysuria, urinary incontinence, and hematuria. The flow sampling methodology is presented in Figure 1.

Patient information, including gender, age, breed, weight, reproductive status, previous history of antimicrobials used, and urinary tract clinical signs, was retrieved from the hospital records. The sample size for the target population was determined using Cochran’s formula [21], employing a 95% confidence level (Z-value = 1.96), a 5% margin of error, and an estimated population proportion (P) of 0.02 (2%) [1]. This proportion was derived from a preliminary study involving 40 cats.

### 2.3. Bacterial Isolation

Feline urine samples collected via cystocentesis were cultured for bacterial isolation at the Veterinary Diagnostic Laboratory, Faculty of Veterinary Medicine, Kasetsart University, Bangkok. A semi-quantitative 1 µL loop was used to inoculate well-mixed urine onto sheep blood agar (Biomedia, Bangkok, Thailand) and MacConkey agar (Difco™, BBL™, BD, Franklin Lakes, NJ, USA) plates. These plates were then incubated at 37 °C for 18–24 h under aerobic conditions. Bacterial growth indicative of a positive culture was assessed and classified as a bacterial infection [22], with a consistent cut-off at 10^3^ CFU/mL, ensuring isolates originated from the urinary bladders of cats with bacterial cystitis. If no bacterial growth was observed after 48 h of incubation, the sample was reported as negative. Bacterial isolates were stored as stock cultures in 20% glycerol at −80 °C.

### 2.4. Bacterial Species Identification

Gram-negative bacteria, excluding *Pseudomonas* spp., were identified using the VITEK^®^ 2 system (Version 9.02.3, BioMérieux, Marcy l’Étoile, France) with the VITEK^®^ 2 GN ID card reference 21341 (BioMérieux, Marcy l’Étoile, France), according to the manufacturer’s instructions. This method provided a confidence level of 90–99% probability. *Pseudomonas* spp. could not be reliably identified to the species level using the VITEK^®^ 2 system and were subsequently characterized using PCR and sequencing targeting the V6 to V9 region (452 bp) [23]. The majority of coagulase-negative *Staphylococcus* species could not be reliably identified to the species level by the VITEK^®^ 2 system. This was due to frequently observed low confidence levels and poor discrimination among isolates using the GP ID card (reference 21342, BioMérieux, Marcy l’Étoile, France), consistent with existing research highlighting inconsistencies in VITEK^®^ 2’s identification of CoNS [24]. Therefore, Gram-positive bacteria were identified using PCR and sequencing targeting the 16S rRNA region (V2 to V6, 974 bp) [25]. DNA was extracted from pure bacterial colonies grown on Tryptic Soy Agar at 37 °C for 18–24 h, utilizing either the boiling method with Tris-EDTA buffer [26] or the E.Z.N.A.^®^ Bacterial DNA Kit (OMEGA Bio-Tek, Norcross, GA, USA).

16S rRNA sequencing for bacterial identification was performed by Macrogen Inc. (Macrogen Inc., Seoul, Republic of Korea). DNA sequences were analyzed using BIOEDIT software (Version 7.2.5), followed by BLASTN alignment. Bacterial species were identified based on BLASTN results, using a minimum query cover and percent identity of 99% [27].

### 2.5. Antimicrobial Susceptibility Testing

Antimicrobial susceptibility testing was performed using the VITEK^®^ 2 system (version 9.02.3) with GN97 and GP81 cards, assessing susceptibility to various antimicrobial agents across 10 distinct drug classes. Quality control for each new card lot was maintained using specific ATCC strains: *Escherichia coli* ATCC 25922 and *Pseudomonas aeruginosa* ATCC 27853 (both from Biomedia, Bangkok, Thailand) for GN97 cards, and *Staphylococcus aureus* ATCC 29213 (DMST 4745, Department of Medical Sciences, Nonthaburi Thailand) and *Enterococcus faecalis* ATCC 29212 (Biomedia, Bangkok, Thailand) for GP81 cards. The VITEK^®^ cards incorporated a comprehensive panel of antimicrobials, including aminoglycosides (amikacin [AMK], gentamicin [GEN], and neomycin [NEO]); penicillins (ampicillin [AMP], benzylpenicillin [BEN], and oxacillin [OXA]); cephalosporins (cephalexin [LEX], cefalotin [CEP], cefotaxime [CTX], cefpodoxime [CPD], cefovecin [VEC], ceftiofur [FUR], cefoxitin [FOX], cefepime [FEP], ceftazidime [CAZ], and clavulanic acid [DA]); carbapenems (imipenem [IPM]); folate pathway inhibitors (sulfamethoxazole/trimethoprim [SXT]); phenicols (chloramphenicol [CMP] and florfenicol [FFC]); fluoroquinolones (enrofloxacin [ENR], marbofloxacin [MAR], and pradofloxacin [PRA]); tetracyclines (doxycycline [DXT], tetracycline [TET], and minocycline [MIN]); nitrofurans (nitrofurantoin [NIT]); lincosamides (clindamycin [CM] and inducible clindamycin resistance [ICR]); and macrolides (erythromycin [ERY]). All breakpoints were interpreted following the Global CLSI-based + Natural Resistance 2017 breakpoint, which serves as the default foundational dataset within the VITEK^®^ 2 system. It is important to note that the VITEK^®^ 2 system did not provide susceptibility data for certain antimicrobials against *Pseudomonas aeruginosa* and *Enterococcus* spp. This absence of data was attributed to either the intrinsic resistance of these organisms to those specific drugs or the drugs not being approved by the FDA for these pathogens. However, even though *Proteus mirabilis* is naturally resistant to nitrofurantoin and tetracycline according to the Performance Standards for Antimicrobial Disk and Dilution Susceptibility Tests for Bacteria Isolated from Animals CLSI VET01 (2023) [28], the VITEK^®^ 2 system still reported susceptibility results for these antimicrobials.

Amoxicillin/clavulanic acid (AMC) susceptibility for *Staphylococcus* species was determined using the Bauer disk diffusion method, applying the Performance Standards for Antimicrobial Disk and Dilution Susceptibility Tests for Bacteria Isolated from Animals (CLSI, 2008) [29], as these results were not available from the VITEK 2^®^ system (version 9.02.3). Similarly, vancomycin susceptibility for *Enterococcus* species was assessed using the Bauer disk diffusion method with the Performance Standards for Antimicrobial Disk and Dilution Susceptibility Tests for Bacteria Isolated from Animals CLSI VET08 (2018) [30].

Extended-spectrum β-lactamase (ESBL) production was screened using the VITEK 2^®^ system. This involved testing with cefepime, cefotaxime, and ceftazidime, which were then compared to the same drugs combined with clavulanic acid, according to the manufacturer’s interpretation criteria.

Antimicrobial susceptibility was classified as low (≤60%), intermediate (61–80%), and high (>80%) [31]. Multidrug resistance (MDR) was defined as resistance or intermediate resistance to at least one agent from three or more antimicrobial classes (≥3 classes), excluding intrinsic resistance [32]. The intrinsic resistance of *Proteus mirabilis* to nitrofurantoin and tetracycline, as previously explained, was not included in this definition.

### 2.6. Definition of Tier 1, 2, and 3 [8,33]

Tier 1: These antimicrobials are the initial treatment choice for sporadic bacterial cystitis, including beta-lactams (e.g., amoxicillin/clavulanic acid) and folate pathway inhibitors (e.g., sulfonamide-trimethoprim).

Tier 2: Their use requires confirmed bacterial resistance to tier 1 antimicrobials supported by antimicrobial susceptibility results; empirical use is generally not recommended. This tier primarily includes fluoroquinolones (e.g., enrofloxacin, marbofloxacin), third-generation cephalosporins (e.g., cefpodoxime, cefovecin), and aminoglycosides (e.g., amikacin, gentamicin). Nitrofurantoin is a specific tier 2 option for sporadic cystitis caused by multidrug-resistant (MDR) pathogens, due to its high urine concentration.

Tier 3: This tier consists of reserved, last-resort agents for critical, life-threatening, multidrug-resistant infections. Their use is strictly contingent upon antimicrobial susceptibility testing results confirming no other tier 1 or 2 options are viable. This tier includes imipenem.

### 2.7. Detection of Methicillin Resistance by mecA Gene PCR

The presence of *mec*A in all *Staphylococcus* isolates (*n* = 24) was determined using conventional PCR, following the protocol described by Kondo et al. (2007) [34]. The *mec*A gene was amplified using the forward primer (mA1) 5′-TGCTATCCACCCTCAAACAGG-3′ and reverse primer (mA2) 5′-AACGTTGTAACCACCCCAAGA-3′, which generates a 286 bp fragment. PCR products were analyzed by 1% agarose gel electrophoresis, stained with ethidium bromide, and visualized under UV light.

### 2.8. Statistical Analysis

The association between positive urine culture results and various risk factors was evaluated using specific statistical tests. Initial analysis of the data confirmed a non-normal distribution (*p*-value ≤ 0.05), as assessed by the One-Sample Kolmogorov–Smirnov Test. Given this non-normal distribution, different statistical approaches were employed based on the nature of the risk factors. For categorical risk factors, such as gender and breed, the Chi-square test was utilized. For continuous risk factors, including age and weight, logistic regression was performed. For all analyses, statistical significance was set at a *p*-value < 0.05, with a 95% confidence level. All statistical analyses were conducted using the STATA software program (version 14) [35].

## 3. Results

### 3.1. Characteristics of Cats with Bacterial Cystitis

A total of 428 cats exhibiting clinical signs consistent with lower urinary tract infection (UTI) were included in the study conducted between June 2022 and May 2023. During this period, 543 urine samples were collected via cystocentesis. Positive bacterial cultures were obtained from 95 cats, indicating a 22.2% prevalence of feline bacterial cystitis. Of these 95 cats, 93 experienced at least one episode of bacterial cystitis, accounting for 108 positive samples. Two cats had confirmed recurrent infections, yielding seven positive cultures over the study period. Table 1 summarizes the data for cats with lower urinary tract signs, categorized by positive and negative bacterial culture results. Among cats diagnosed with bacterial cystitis, neutered males, domestic shorthair cats, young adults, and those weighing less than 6.8 kg were most commonly represented. However, no significant correlation was found between the examined risk factors (gender, sterilization status, breed, age, and weight) and positive urine culture results.

### 3.2. Feline Uropathogenic Bacterial Species

The 115 positive urine samples yielded 125 bacterial isolates. Of these, 105 (91.3%) indicated single infections, while 10 (8.7%) showed mixed infections. Table 2 presents the bacterial species associated with feline urinary tract infections (UTIs) and their prevalence*. Escherichia coli* (*n* = 31, 24.8%) was the most frequently isolated bacterium, followed by *Staphylococcus* spp. (*n* = 24, 19.2%), *Proteus mirabilis* (*n* = 17, 13.6%), *Pseudomonas aeruginosa* (*n* = 15, 12%), and *Enterococcus* spp. (*n* = 15, 12%). Other Gram-negative bacteria included *Klebsiella pneumoniae* (*n* = 8, 6.4%), *Enterobacter cloacae* complex (*n* = 6, 4.8%), *Acinetobacter baumannii* complex (*n* = 3, 2.4%), and single isolates of *Aeromonas* sp. and *Enterobacter aerogenes*. Among the *Staphylococcus* species, isolates included *Staphylococcus felis* (*n* = 11, 8.8%), *Staphylococcus pseudintermedius* (*n* = 7, 5.6%), *Staphylococcus epidermidis* (*n* = 4, 3.2%)*, Staphylococcus aureus* (*n* = 1, 0.8%), and *Staphylococcus schleiferi* (*n* = 1; 0.8%). Additional Gram-positive bacteria isolates were *Enterococcus faecium* (*n* = 7, 5.6%), *Enterococcus faecalis* (*n* = 6, 4.8%), and single isolates of *Enterococcus avium*, *Enterococcus raffinosus*, *Lactococcus garvieae*, *Streptococcus canis*, and *Corynebacterium urealyticum* (each *n* = 1, 0.8%). In two cases of recurrent cystitis, different bacterial species were isolated from multiple urine samples collected over time. Four samples were retrieved from the first cat over 12 months, which yielded *Enterobacter cloacae* (Sample 1), a mixed infection of *Klebsiella pneumoniae* and *Lactococcus garvieae* (Sample 2), *Klebsiella pneumoniae* (Sample 3), and a recurrence of *Enterobacter cloacae* (Sample 4). Three samples were retrieved from the second cat during the same period, with a mixed infection of *Enterobacter cloacae* and *Klebsiella pneumoniae* (Sample 1), *Enterococcus avium* (Sample 2), and *Proteus mirabilis* (Sample 3).

### 3.3. Antimicrobial Susceptibility Profiles

Table 2 also presents varying rates of multidrug resistance (MDR) among the tested bacterial isolates, with MDR observed in 83 out of 120 isolates (69.2%). A 100% MDR rate was exhibited by *Enterococcus* spp., *Klebsiella pneumoniae*, *Staphylococcus epidermidis*, *Acinetobacter baumannii complex*, and *Staphylococcus schleiferi*. The Gram-positive bacterium *Staphylococcus pseudintermedius* also showed a high MDR rate of 85.7%. Among other Gram-negative bacteria, high MDR rates (>50%) were observed in *Escherichia coli* (67.7%), *Proteus mirabilis* (76.5%), and the *Enterobacter cloacae* complex (83.3%). In contrast, *Staphylococcus felis* (18.2%) and *Pseudomonas aeruginosa* (33.3%) exhibited lower MDR frequencies.

Heatmaps and percentages of antimicrobial susceptibility for the most common Gram-negative pathogens are presented in Figure 2, Figure 3 and Figure 4. *Escherichia coli* displayed low susceptibility to tier 1 agents: ampicillin, sulfamethoxazole/trimethoprim, and amoxicillin/clavulanic acid (Figure 2). For tier 2 and 3 agents, *Escherichia coli* demonstrated variable susceptibility. Specifically, low susceptibility was observed for fluoroquinolones, tetracyclines, and gentamicin. *Escherichia coli* showed intermediate susceptibility to most cephalosporins. However*, Escherichia coli* demonstrated very high susceptibility (>90%) to other tier 2 and 3 agents. These susceptibility patterns align with the observation that MDR *Escherichia coli* isolates exhibited a core resistance phenotype, characterized by co-resistance to penicillin, cephalosporins, and fluoroquinolones, often with additional sulfamethoxazole/trimethoprim or tetracycline resistance. Furthermore, extended-spectrum beta-lactamase (ESBL) screening in *Escherichia coli* revealed a positive rate of 19.3% (6 of 31 isolates), with all positive isolates found to be MDR.

*Proteus mirabilis* exhibited low susceptibility to multiple antimicrobials (Figure 3). Among tier 1 agents, susceptibility rates were low. Similarly, reduced susceptibility was observed among tier 2 and 3 agents. Notably, *Proteus mirabilis* demonstrated complete resistance to nitrofurantoin and tetracycline, which is consistent with its known intrinsic resistance to these agents. Intermediate susceptibility was observed for third-generation cephalosporins. In contrast, high susceptibility (>80%) was observed for amikacin and neomycin. Six of the thirteen MDR *Proteus mirabilis* isolates were resistant to penicillin, first-generation cephalosporins, and fluoroquinolones.

*Pseudomonas aeruginosa* exhibited low susceptibility to tier 2 agents, showing complete resistance to third-generation cephalosporins and low resistance to fluoroquinolones (Figure 4). Intermediate susceptibility was observed for gentamicin. However, high susceptibility was retained to other antimicrobials, particularly tier 2 amikacin and tier 3 imipenem.

Other Gram-negative bacteria, including *Klebsiella pneumoniae*, *Enterobacter cloacae* complex, and *Acinetobacter baumannii* complex, showed low susceptibility to tier 1 and tier 2 agents. Conversely, they retained high susceptibility to tier 3 agents, specifically imipenem, with the notable exception of *Acinetobacter baumannii* complex, which showed lower susceptibility to imipenem. ESBL screening in *Klebsiella pneumoniae* revealed 2/8 positive isolates.

Figure 5 illustrates the antimicrobial susceptibility profiles of two coagulase-negative *Staphylococcus* species (CoNS): *Staphylococcus felis* and *Staphylococcus epidermidis*. *Staphylococcus felis*, the predominant species identified, showed high susceptibility to most antimicrobial agents. In contrast, *Staphylococcus epidermidis* exhibited complete resistance to benzylpenicillin and the tier 2 agents cefovecin and clindamycin. It displayed intermediate susceptibility to the tier 1 amoxicillin/clavulanic acid and the tier 2 erythromycin and gentamicin. However, *Staphylococcus epidermidis* was highly susceptible to the tier 1 agent sulfamethoxazole/trimethoprim and several tier 2 agents. Regarding methicillin resistance, the *mec*A gene was identified in only one of the *Staphylococcus felis* isolates, compared to a higher number in *Staphylococcus epidermidis* isolates. Interestingly, this *mec*A-positive *Staphylococcus felis* isolate did not show resistance to any antimicrobials. Conversely, the high proportion of methicillin-resistant *Staphylococcus epidermidis* isolates corresponded to a high MDR percentage within that species.

Figure 6 presents the antimicrobial susceptibility profiles of three coagulase-positive *Staphylococcus* species (CoPS): *Staphylococcus pseudintermedius*, *Staphylococcus schleiferi*, and *Staphylococcus aureus*. *Staphylococcus pseudintermedius*, the most frequently identified species among these CoPS, exhibited low to intermediate susceptibility to the tier 1 agents. For tier 2 agents, their susceptibility varied, demonstrating low susceptibility to several, yet high susceptibility to others within this tier. Notably, methicillin-resistant *Staphylococcus pseudintermedius* (MRSP) was identified in a high proportion of isolates. The *mec*A gene was also detected in one *Staphylococcus schleiferi* isolate, though it was absent in *Staphylococcus aureus*. All MRSP isolates were MDR, frequently exhibiting co-resistance to the aforementioned drugs. Overall, the *mec*A gene, which indicates methicillin resistance, was detected in a notable proportion of *Staphylococcus* isolates.

Figure 7 presents the antimicrobial susceptibility profiles of four *Enterococcus* species: *Enterococcus faecalis*, *Enterococcus faecium*, *Enterococcus avium*, and *Enterococcus raffinosus*. *Enterococcus faecalis* showed high susceptibility to tier 1 agents. However, it exhibited low susceptibility to several tier 2 agents, with intermediate susceptibility observed for chloramphenicol. High susceptibility was also noted for florfenicol, nitrofurantoin, and the restricted drug vancomycin. In contrast, *Enterococcus faecium* revealed low susceptibility to various tier 1 and tier 2 agents, though it similarly demonstrated high susceptibility to florfenicol and vancomycin. Notably, all *Enterococcus* isolates were MDR, with varying susceptibility profiles across the different species.

Complete susceptibility percentages for all Gram-negative and Gram-positive bacterial species (Appendix A) are provided in the Appendix A, alongside a broad summary of the overall antimicrobial resistance profiles (Appendix A). This data illustrates general patterns of how UTI bacteria respond to various antimicrobials, underscoring the importance of selecting treatments based on their specific resistance profiles.

## 4. Discussion

This study investigated the prevalence of bacterial species and their antimicrobial resistance in cats with urinary tract infections (UTIs) treated at the Veterinary Teaching Hospital of Kasetsart University, Bangkok, Thailand. Over a one-year period (2022–2023), 21.2% (115/543) of urine samples from 22.2% (95/428) of the cats yielded positive bacterial cultures, while 77.8% (333/428) had no bacterial growth. The proportion of bacterial UTI is relatively higher compared to previous American reports (3%) [36], but closer to European findings (25%) [4]. This discrepancy may be due to the study population, which consisted of a large number of primary hospitals and cats referred to a tertiary care hospital.

The high proportion (77.8%) of urine samples from cats showing no bacterial growth can be attributed to several factors. Firstly, feline lower urinary tract disease (FLUTD) encompasses various conditions beyond bacterial cystitis, which can present with similar FLUTD clinical signs [4]. Secondly, prior antimicrobial use or potential treatment at other clinics may have suppressed bacterial growth. Additionally, the presence of organisms that do not survive under standard aerobic urine culture conditions, such as anaerobes and *Mycoplasma* spp., could also explain the lack of bacterial growth [37]. Therefore, it is important to consider these factors when interpreting negative urine culture results in cats, particularly those with a history of antimicrobial therapy or recent veterinary care.

Statistical analysis revealed no significant association between the evaluated variables and positive urine cultures in the 95 cats studied. However, neutered males were more commonly affected than neutered females. Most of the cats were adults, with a mean age of 6.3 years, and domestic shorthair (DSH) was the most frequently affected breed. This finding is consistent with local studies [12]. In contrast, a study conducted in the United States reported that spayed females and Abyssinian cats were significantly associated with bacterial urinary tract infections [36], reflecting regional differences in cat breeds and management practices. The observed variability suggests that in areas where pedigree cats are more common, these breeds may be more frequently affected.

*Escherichia coli* (24.8%), *Staphylococcus* spp. (19.2%), *Proteus mirabilis* (13.6%), and both *Pseudomonas aeruginosa* and *Enterococcus* spp. (12% each) were the most common pathogens identified in this study. Previous studies have consistently reported *Escherichia coli* as the most frequently detected pathogen in the United States [38], Italy [39], Germany [40], and Europe [9]. Furthermore, these studies often report *Enterococcus* spp. as the second most common pathogen, followed by *Proteus mirabilis* and *Staphylococcus* spp., depending on the region. Interestingly, a previous study in Chiang Mai, Thailand [12], identified *Pseudomonas aeruginosa* as the most common bacterial isolate (25%). While the current study also observed *Pseudomonas aeruginosa* at a prevalence of 12%, this finding contrasts with the lower prevalence rates reported in the previously mentioned countries (2.7% to 6.5%). This discrepancy may be attributed to environmental and climatic factors, as *Pseudomonas aeruginosa* is a non-enteric opportunistic pathogen commonly found in soil and water sources [41]. Additionally, Thailand’s high humidity may contribute to the increased prevalence of *Pseudomonas aeruginosa* [42].

Feline uropathogenic *Staphylococcus* species were identified using 16S rRNA PCR sequencing, providing species differentiation, a detail infrequently reported in Thailand. Of these, *Staphylococcus felis* was the most prevalent *Staphylococcus* species causing UTIs in cats, in concordance with findings from studies conducted elsewhere (e.g., Australia and Belgium) [9,43]. This is consistent with the suggestion by Olin & Bartges (2015) that cats may harbor unique strains of *Staphylococcus felis* that cannot be reliably differentiated from other coagulase-negative *Staphylococcus* species using commercial phenotypic identification methods [44]. Additionally, less commonly reported species identified using 16S rRNA PCR sequencing included *Corynebacterium urealyticum*, previously associated with urethral obstruction in cats [45], and *Lactococcus garvieae*, which has been linked to UTIs in humans [46]. The infrequent reporting of these species, both in the general literature and specifically within Thailand, may be attributed to earlier limitations in bacterial identification methods. However, both *Corynebacterium urealyticum* and *Lactococcus garvieae* have been identified in feline urine samples in studies from other countries [47,48,49].

Based on this study, an alarming trend of antimicrobial resistance is evident in the predominant bacteria causing UTIs in Thailand. *Escherichia coli* exhibits low susceptibility to tier 1 and various tier 2 antimicrobials, including penicillin, sulfamethoxazole/trimethoprim, first-generation cephalosporins, fluoroquinolones, and tetracyclines. This aligns with a 2021 Chiang Mai study by Amphaiphan et al. (2021) [12], which reported similarly low susceptibility (0–40%) to ampicillin, sulfamethoxazole/trimethoprim, cephalexin, enrofloxacin, and norfloxacin, suggesting a widespread issue across the country. However, a notable difference in the Chiang Mai study was the high susceptibility (80%) of *Escherichia coli* to amoxicillin/clavulanic acid, which they suggested could still be used as an empirical choice [12]. Particularly concerning is the low susceptibility to fluoroquinolones, with resistance rates of 64.5% for enrofloxacin, 54.8% for marbofloxacin, and 58% for pradofloxacin. In striking contrast, D’Août et al. (2022) reported 100% susceptibility of *Escherichia coli* to fluoroquinolones in the United Kingdom [10], highlighting a significant geographical disparity. This resistance has important implications, as fluoroquinolones are crucial for treating UTIs and managing multidrug-resistant infections in both human and veterinary medicine [39]. Furthermore, the study revealed a high rate of MDR in *Escherichia coli* (67.7%), characterized by core co-resistance to penicillin, cephalosporins, and fluoroquinolones. This rate is notably high, even when compared to human data in Thailand, where *Escherichia coli* from human UTI patients showed an 87% MDR rate and similar resistance to quinolones [50]. However, *Escherichia coli* still demonstrated high susceptibility to certain tier 2 drugs, including amikacin (93.5%), neomycin (90.3%), and nitrofurantoin (90.3%), as well as the tier 3 drug imipenem (96.8%).

Among other Gram-negative bacteria, *Proteus mirabilis* and *Pseudomonas aeruginosa* were the next most common species identified in this study. Consistent with research by Amphaiphan et al. [12] in Thailand, which reported *Proteus mirabilis* had low susceptibility (0–50%) to ampicillin, cephalexin, gentamicin, and fluoroquinolones (e.g., enrofloxacin and norfloxacin), the current study also found *Proteus mirabilis* exhibited low susceptibility (<50%) to ampicillin, cephalexin, and fluoroquinolones. Furthermore, major (7/13, 53.8%) of MDR *Proteus mirabilis* isolates also demonstrated resistance to fluoroquinolones. This may be due to acquired resistance (mutations) and a potentially increasing prevalence of resistant strains, as highlighted by international reports [39,51,52]. *Pseudomonas aeruginosa* naturally resists many antimicrobials, and these findings for it mirrored international trends, showing high resistance to cephalosporins and fluoroquinolones [9,39].

This study identified *Klebsiella pneumoniae* and *Enterobacter cloacae* complex in recurrent feline infections (reinfection and relapse), exhibiting concerning MDR rates of 100% and 83.3%, respectively. While *Escherichia coli* was the most frequently isolated pathogen, with a 67.7% MDR rate, it was notably absent in these recurrent cases. This contrasts with a 66.7% *Escherichia coli* recurrence rate reported in a US study [38]. To understand this discrepancy and to further characterize the high MDR observed in the *Klebsiella pneumoniae* and *Enterobacter cloacae* complex, particularly their association with recurrence, strain, and virulence, AMR gene profiling is highly recommended. This would enable a detailed profile of these pathogens, identifying potential clonal spread, clarifying recurrence mechanisms, and elucidating underlying resistance mechanisms. While acknowledging the importance of patient factors, a detailed analysis of such associations was beyond the scope of this study due to the limitations of available retrospective data. Nevertheless, further research in this area is crucial.

This research emphasizes the critical need for targeted antimicrobial therapy based on susceptibility testing, particularly for Gram-positive bacteria, feline pathogens like *Staphylococcus* spp. and *Enterococcus* spp. This is due to the significant differences observed in resistance patterns among individual species. For *Staphylococcus* spp., *Staphylococcus felis* generally showed high susceptibility to all antimicrobials tested. However, its decreased susceptibility to benzylpenicillin suggests caution when selecting beta-lactam antimicrobials. In contrast, *Staphylococcus pseudintermedius* exhibited significantly higher resistance to fluoroquinolones, a widely used class of antimicrobials for treating bacterial infections. Its low susceptibility in this class (14.3%) warrants careful consideration. Nevertheless, *Staphylococcus pseudintermedius* showed higher susceptibility to aminoglycosides such as gentamicin and amikacin, indicating these as more reliable therapeutic options. Furthermore, distinct resistance patterns were found between *Enterococcus faecium* and *Enterococcus faecalis*. Notably, *Enterococcus faecium* displayed higher resistance to benzylpenicillin, amoxicillin/clavulanic acid, and nitrofurantoin compared to *Enterococcus faecalis*. This trend is consistent with studies in the United States, where *Enterococcus faecium* tends to show higher resistance to several antimicrobials compared to *Enterococcus faecalis* [53]. Overall, the diverse resistance profiles among these Gram-positive species underscore the paramount importance of species-level identification and susceptibility testing to ensure the most effective and appropriate treatment for the specific bacterial pathogen.

The International Society for Companion Animal Infectious Disease (ISCAID) guidelines emphasize that optimal empirical antimicrobial choice depends on local pathogen and resistance patterns [8,22]. While ISCAID recommends tier 1 antimicrobials like amoxicillin/clavulanate and sulfamethoxazole/trimethoprim for bacterial cystitis, study susceptibility results show limitations for their empirical use. The recommended susceptibility rate for reliable empirical options is over 80% [31,54]. In contrast, the main pathogens had significantly lower susceptibility: *Escherichia coli* (amoxicillin/clavulanate 54.8%, sulfamethoxazole/trimethoprim 48.4%), *Proteus mirabilis* (amoxicillin/clavulanate 29.4%, sulfamethoxazole/trimethoprim 47.1%), and *Staphylococcus* spp., particularly *Staphylococcus pseudintermedius* (amoxicillin/clavulanate 42.9%, sulfamethoxazole/trimethoprim 71.4%). This indicates restrictions in empirical drug selection.

When tier 1 drugs are not suitable, ISCAID guidelines suggest considering nitrofurantoin, especially in cases involving multidrug-resistant infections. Other options include fluoroquinolones (e.g., enrofloxacin, marbofloxacin, and pradofloxacin) and third-generation cephalosporins (e.g., cefovecin and ceftiofur). However, the European Medicines Agency (EMA) classifies these three drug groups as Highest Priority Critically Important Antimicrobials (HPCIAs), restricting their veterinary use due to their critical importance in treating multidrug-resistant human infections [55]. Study susceptibility results reveal high resistance to these alternatives. Susceptibility to third-generation cephalosporins and fluoroquinolones was below 80% in the main pathogens (*Escherichia coli*, *Proteus mirabilis*, and *Staphylococcus pseudintermedius*). While nitrofurantoin still showed high susceptibility (>80%) in most key isolates, it is ineffective against *Proteus mirabilis* and *Pseudomonas aeruginosa*, which are intrinsically resistant. Despite being an option, international reports [56] raise concerns about its potential side effects (including gastrointestinal issues), toxicity, and poor pharmacokinetics, often requiring injections that need owner cooperation. Therefore, nitrofurantoin should preferably be reserved for MDR cases or when susceptibility testing confirms its effectiveness against the specific pathogen.

The high resistance seen in the aforementioned drug classes might be due to their frequent use as first-line treatments [38], reported usage of critical antimicrobials in cats, and issues with over-prescription in felines [40], all of which could contribute to increased resistance [44]. It is crucial to maintain continuous surveillance of resistance patterns for both tier 1 and tier 2 antimicrobials to ensure effective treatment options remain available.

Beyond the challenges of AMR in veterinary medicine, there is growing concern about zoonotic transmission and the emergence of MDR bacteria. This study found a high prevalence of resistance among common isolates: specifically, 41.7% of *Staphylococcus* isolates were methicillin-resistant, and 19.3% of *Escherichia coli* screened positive for ESBL production, both exhibiting concerning multidrug-resistant characteristics. These findings are particularly relevant given the documented presence of similar MDR bacterial strains in both animals and humans, indicating a significant potential for inter-species sharing of resistance genes—a trend observed globally and locally. For instance, a Thai study [57] found that antimicrobial-resistant *Escherichia coli* isolates from veterinarians, pet owners, and pets often carried shared ESBL genes (*bla*_CTX-M, *bla*_TEM). The high *mec*A gene detection in *Staphylococcus pseudintermedius* observed in the current study further emphasizes zoonotic concern, as specific MRSP strains (e.g., ST 45, ST 181) have been identified in both dogs and their human contacts (owners and veterinarians) [58]. Further supporting this, a Portuguese report on *Proteus mirabilis* revealed that a substantial portion of its clusters contained highly genetically similar strains from both pets and humans, with some showing complete similarity (e.g., dog to human community isolate) [59]. These findings underscore the strong link between human and animal health and the critical need for comprehensive AMR strategies, as resistance is often acquired through horizontal gene transfer from external sources, though chromosomal mutations and selective pressure from antimicrobials also play significant roles [60].

Previous studies have highlighted the global distribution of methicillin-susceptible *Staphylococcus* strains that detected the *mec*A gene, with similar occurrences also noted in Thailand [19,61]. Further supporting this trend, our study identified three isolates—*Staphylococcus pseudintermedius*, *Staphylococcus schleiferi*, and *Staphylococcus felis*—that were methicillin(oxacillin)-susceptible despite being positive for the *mec*A gene. This reveals a discordance between the detection of the *mec*A gene and phenotypic methicillin (oxacillin) resistance. This discordance may be associated with *mec*A gene expression regulation involving *mec*R1 (an inducer) and *mec*I (a repressor), while *bla* regulators, such as *bla*R1–*bla*I, may also regulate *mec* gene expression in specific types of the *mec* gene complex [62].

The primary limitation of this study is its retrospective design, arising from a reliance on anonymized outpatient (OPD) data collected solely for routine veterinary diagnostics. This inherent design meant the data was not structured to capture detailed clinical outcomes, such as patient recovery, re-hospitalization rates, or length of hospital stays. Similarly, comprehensive patient information, including chronic illness details beyond initial comorbidities like chronic kidney disease and urethral stenosis, was incompletely recorded. Of particular note, specific risk factors and precise antimicrobial usage data were not consistently captured retrospectively, falling outside the scope of the available dataset. Nevertheless, these identified limitations highlight critical areas for more comprehensive prospective studies in the future.

## 5. Conclusions

This study identified the most common bacterial pathogens causing feline urinary tract infections in Thailand. Antimicrobial susceptibility testing of the predominant pathogens revealed high resistance to commonly used first- and second-line antimicrobials, including amoxicillin/clavulanic acid, sulfamethoxazole/trimethoprim, and fluoroquinolones. Furthermore, MDR was observed in both Gram-negative and Gram-positive bacteria frequently implicated in feline UTIs. This highlights the limitations of using tier 1 antimicrobials for empirical treatment and underscores the importance of both bacterial species identification and antimicrobial susceptibility testing before administering treatment. Consequently, the local susceptibility data from this study can guide the appropriate selection of antimicrobials for treating feline bacterial urinary tract infections. Therefore, regular monitoring and surveillance of AMR data at the local level are crucial for promptly detecting undesirable trends.

## Figures and Tables

**Figure 1 animals-15-02235-f001:**
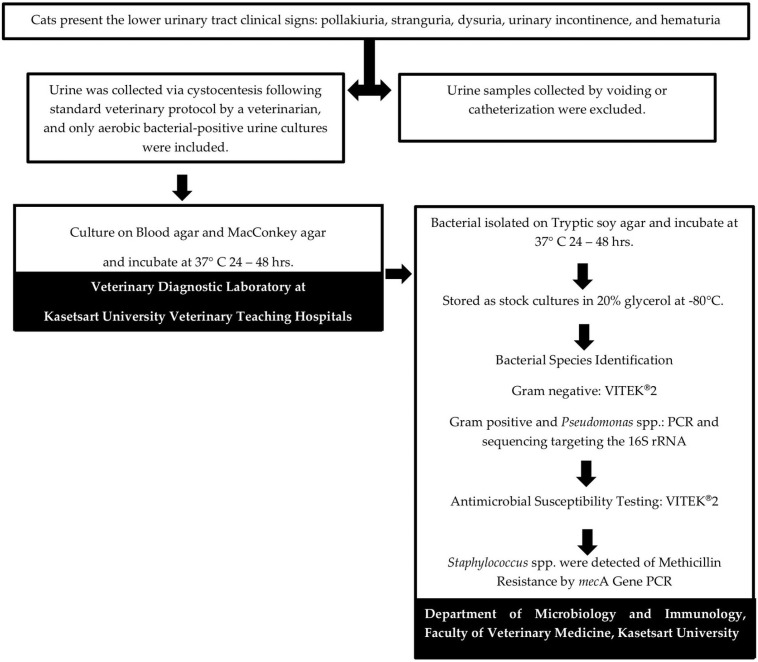
Workflow illustrating the reception of clinical samples from the diagnostic hospital, followed by the research process of isolating and analyzing bacteria from leftover diagnostic materials.

**Figure 2 animals-15-02235-f002:**
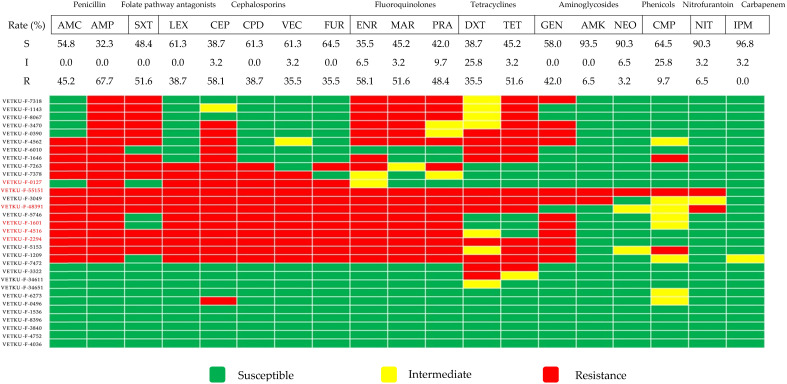
Heatmap of *Escherichia coli* antimicrobial susceptibility profiles and calculated susceptibility/resistance percentages (*n* = 31). Each coded row corresponds to a single isolate, with red-colored labels specifically indicating isolates positive for ESBL screening. Abbreviations used are AMC, amoxicillin/clavulanic acid; AMP, ampicillin; SXT, sulfamethoxazole/trimethoprim; LEX, cephalexin; CEP, cephalothin; CPD, cefpodoxime; VEC, cefovecin; FUR, ceftiofur; ENR, enrofloxacin; MAR, marbofloxacin; PRA, pradofloxacin; DXT, doxycycline; TET, tetracycline; GEN, gentamicin; AMK, amikacin; NEO, neomycin; CMP, chloramphenicol; NIT, nitrofurantoin; IPM, imipenem.

**Figure 3 animals-15-02235-f003:**
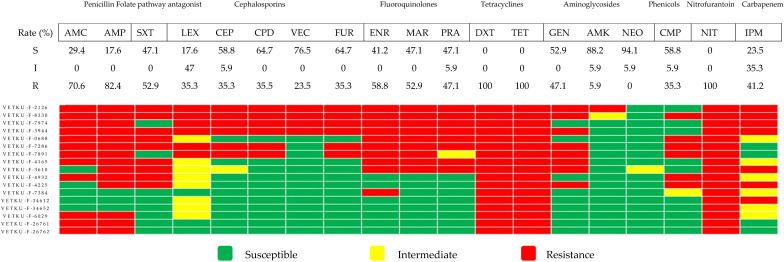
Heatmap of *Proteus mirabilis* antimicrobial susceptibility profiles and calculated susceptibility/resistance percentages (*n* = 17). Each coded row corresponds to a single isolate. Abbreviations used are AMC, amoxicillin/clavulanic acid; AMP, ampicillin; SXT, sulfamethoxazole/trimethoprim; LEX, cephalexin; CEP, cephalotin; CPD, cepodoxime; VEC, cefovecin; FUR, ceftiofur; ENR, enrofloxacin; MAR, marbofloxacin; PRA, pradofloxacin; DXT, doxycycline; TET, tetracycline; GEN, gentamicin; AMK, amikacin; NEO, neomycin; CMP, chloramphenicol; IPM, imipenem.

**Figure 4 animals-15-02235-f004:**
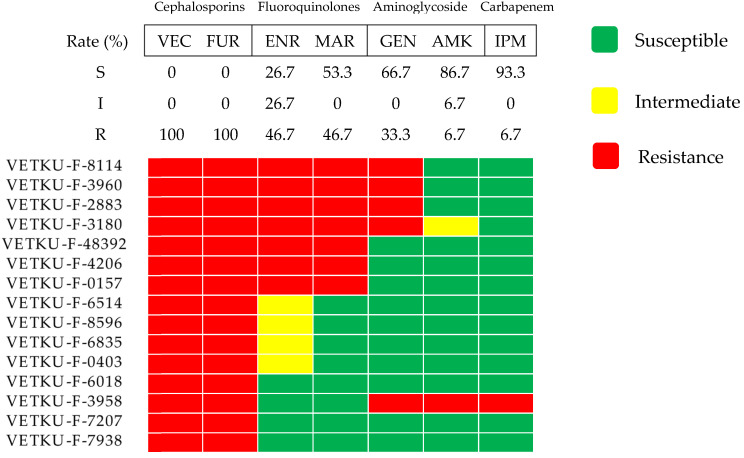
Heatmap of *Pseudomonas aeruginosa* antimicrobial susceptibility profiles and calculated susceptibility/resistance percentages (*n* = 15). Each coded row corresponds to a single isolate. Abbreviations used are VEC, cefovecin; FUR, ceftiofur; ENR, enrofloxacin; MAR, marbofloxacin; GEN, gentamicin; AMK, amikacin; IPM, imipenem.

**Figure 5 animals-15-02235-f005:**
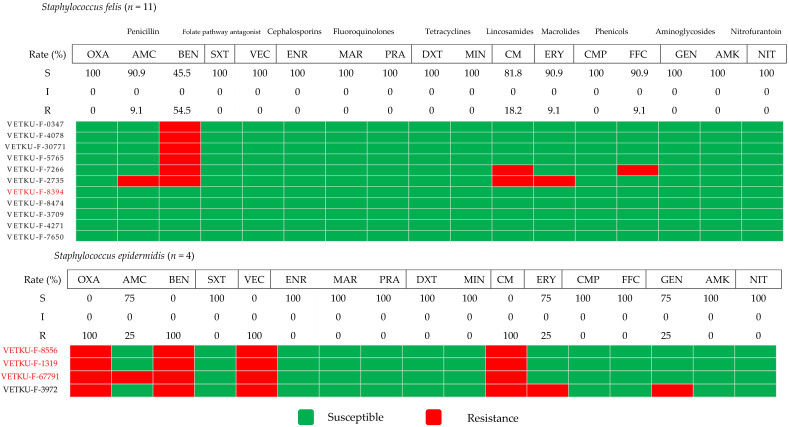
Heatmap of coagulase-negative *Staphylococcus* species antimicrobial susceptibility profiles and calculated susceptibility/resistance percentages (*n* = 15). Each coded row corresponds to a single isolate, with red-colored labels specifically indicating isolates positive for the *mecA* gene. Abbreviations used are OXA, oxacillin; AMC, amoxicillin/clavulanic acid; BEN, benzylpenicillin; SXT, sulfamethoxazole/trimethoprim; VEC, cefovecin; ENR, enrofloxacin; MAR, marbofloxacin; PRA, pradofloxacin; DXT, doxycycline; MIN, minocycline; CM, clindamycin; ERY, erythromycin; CMP, chloramphenicol; FFC, florfenicol; GEN, gentamicin; AMK, amikacin; NIT, nitrofurantoin.

**Figure 6 animals-15-02235-f006:**
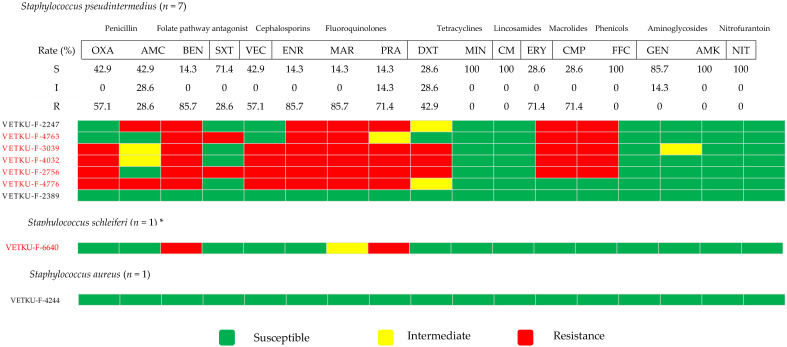
Heatmap of coagulase-positive *Staphylococcus* species antimicrobial susceptibility profiles and calculated susceptibility/resistance percentages (*n* = 9). Each coded row corresponds to a single isolate, with red-colored labels specifically indicating isolates positive for the *mecA* gene. Abbreviations used are OXA, oxacillin; AMC, amoxicillin/clavulanic acid; BEN, benzylpenicillin; SXT, sulfamethoxazole/trimethoprim; VEC, cefovecin; ENR, enrofloxacin; MAR, marbofloxacin; PRA, pradofloxacin; DXT, doxycycline; MIN, minocycline; CM, clindamycin; ERY, erythromycin; CMP, chloramphenicol; FFC, florfenicol; GEN, gentamicin; AMK, amikacin; NIT, nitrofurantoin. * It was predicted as a coagulase-positive *Staphylococcus* based on hemolysis.

**Figure 7 animals-15-02235-f007:**
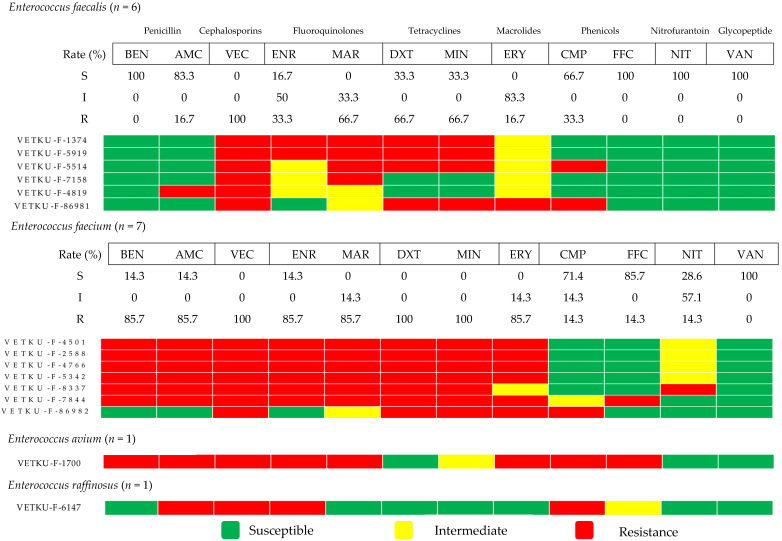
Heatmap of *Enterococcus* species antimicrobial susceptibility profiles and calculated susceptibility/resistance percentages (*n* = 13). Each coded row corresponds to a single isolate. BEN, benzylpenicillin; AMC, amoxicillin/clavulanic acid; VEC, cefovecin; ENR, enrofloxacin; MAR, marbofloxacin; DXT, doxycycline; MIN, minocycline; ERY, erythromycin; CMP, chloramphenicol; FFC, florfenicol; NIT, nitrofurantoin; VAN, vancomycin.

**Table 1 animals-15-02235-t001:** Demographic characteristics of cats with lower urinary tract signs by bacterial culture status.

Variables	Cats with Feline Lower Urinary Tract Clinical Signs, *n* = 428 (%)	
	No. of Cats with Bacterial Cystitis *n* = 95	No. of Cats with No Bacterial Growth *n* = 333
Gender		
Male	68 (71.6)	250 (75.1)
Sexually intact male	10 (10.5)	69 (20.7)
Castrated male	45 (47.4)	128 (38.4)
Sterilization status unknown	13 (13.6)	53 (15.9)
Female	27 (28.4)	83 (24.9)
Sexually intact female	3 (3.2)	10 (3)
Spayed female	19 (20.0)	55 (16.5)
Sterilization status unknown	5 (5.3)	18 (5.4)
Age group		
Kitten (Birth–1 year)	4 (4.2)	17 (5.1)
Young adult (1–6 years)	47 (49.5)	181 (54.4)
Mature adult (7–10 years)	24 (25.3)	75 (22.5)
Senior (10 years+)	20 (21.0)	60 (18.0)
Weight		
<6.8 kg.	77 (81)	267 (80.2)
≥6.8 kg.	3 (3.2)	20 (6)
Not specified	15 (15.8)	46 (13.8)
Breed		
Domestic shorthair	67 (70.5)	214 (64.3)
Persian	13 (13.7)	59 (17.7)
Scottish fold	6 (6.3)	29 (8.7)
American Wirehair	4 (4.2)	10 (3)
Others	5 (5.3)	21 (6.3)

**Table 2 animals-15-02235-t002:** Prevalence of Uropathogenic Bacteria in Feline Urinary Tract Infections.

	Bacterial Species Identification *n* = 125		Multi-Drug Resistance (MDR) *n* = 120 *	
	No. of Isolates	Percentage	No. of Isolates	Percentage
Gram-negative	82	65.6	55	45.8
*Escherichia coli*	31	24.8	21	67.7
*Proteus mirabilis*	17	13.6	13	76.5
*Pseudomonas aeruginosa*	15	12	5	33.3
*Klebsiella pneumoniae*	8	6.4	8	100
*Enterobacter cloacae* complex	6	4.8	5	83.3
*Acinetobacter baumannii* complex	3	2.4	3	100
*Aeromonas* spp.	1	0.8	-	-
*Enterobacter aerogenes*	1	0.8	0	0
Gram-positive	43	33.6	28	23.3
*Staphylococcus* spp.	24	19.2	13	54.2
*Staphylococcus felis*	11	8.8	2	18.2
*Staphylococcus pseudintemedius*	7	5.6	6	85.7
*Staphylococcus epidermidis*	4	3.2	4	100
*Staphylococcus aureus*	1	0.8	0	0
*Staphylococcus schleiferi*	1	0.8	1	100
*Enterococcus* spp.	15	12	15	100
*Enterococcus faecium*	7	5.6	7	100
*Enterococcus faecalis*	6	4.8	6	100
*Enterococcus avium*	1	0.8	1	100
*Enterococcus raffinosus*	1	0.8	1	100
Others	4	3	-	-
*Lactococcus garvieae*	2	1.6	-	-
*Corynebacterium urealyticum*	1	0.8	-	-
* Streptococcus canis*	1	0.8	-	-
Total	125	100	83	69.2

* Antimicrobial susceptibility was determined for 120 of 125 bacterial isolates using the automated VITEK^®^ 2 system, supplemented by disk diffusion where necessary. MDR analysis was performed on these 120 isolates. The remaining five pathogens were not tested for susceptibility, as the VITEK^®^ 2 GP81 and GN97 AST cards are not claimed or validated by the manufacturer for use with these specific organisms.

## Data Availability

The data presented in this study are available on request from the corresponding author.

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
