# Peer review of "Investigation of Bacterial Species and Their Antimicrobial Drug Resistance Profile in Feline Urinary Tract Infection in Thailand"

_animals, 2025, doi:10.3390/ani15152235_

Round 1
Reviewer 1 Report
Comments and Suggestions for Authors
This study investigated the prevalence of bacterial species and their antimicrobial resistance in cats with urinary tract infections (UTIs) treated at the Veterinary Teaching Hospital of Kasetsart University, Bangkok, Thailand. Among the cats tested, 115 (21.2%) were culture-positive for bacteria, resulting in a diagnosis of urinary tract infection (UTI) in 95 cats (22.2%). The most common bacterial isolates were Escherichia coli (24.8%), Staphylococcus spp. (19.2%), Proteus mirabilis (13.6%), Pseudomonas aeruginosa (12.0%), and Enterococcus spp. (12.0%). Among the Staphylococcus isolates, S. felis (8.8%) and S. pseudintermedius (5.6%) were the most frequently identified species. Antimicrobial testing revealed alarming resistance, with 69.2% of isolates exhibiting multidrug resistance (MDR).
The manuscript presents an interesting and significant topic commendable in context feline urinary tract infections caused by bacteria and their antimicrobials profile that can lead to better treatment decision. The result is clearly stated and the tables and figures are well presented. The conclusion is consistent with the evidence presented in the results, and properly addressing the study question. However, the following revisions must address before considering it for publication
----------------------------------------------------------------------------------------------------------
Comment for authors
- No quality control was used or mentioned during the performance of the antimicrobial susceptibility testing (AST), leading to the assumption that the test was conducted without a positive control.
- Line 206: Please add P- value to statistical analysis
- Please ensure consistency in the use of the bacterial name throughout the manuscript. Use either Escherichia coli or E. coli instantly, but not both interchangeably within the text. The same with other bacterial species.
- Line 129: please provide the date and purpose of the study in the ethical approval section. Also, please add the consent information
Author Response
Response to Reviewer 1's comments
Comments 1: No quality control was used or mentioned during the performance of the antimicrobial susceptibility testing (AST), leading to the assumption that the test was conducted without a positive control.
Response 1: Thank you for your valuable feedback on our manuscript. Regarding antimicrobial susceptibility testing (AST), the VITEK® 2 system incorporates rigorous quality control measures, with each card featuring a positive control well (containing no antimicrobial agent) to ensure organism viability and growth.
We perform quality control (QC) for each VITEK® 2 card lot, specifically when the card lot changes, using standardized ATCC microbial strains as recommended by the card's package insert.
For Gram-negative AST (GN97), Escherichia coli ATCC 25922 and Pseudomonas aeruginosa ATCC 27853 are used. For Gram-positive AST (GP81), Staphylococcus aureus ATCC 29213 and Enterococcus faecalis ATCC 29212 are used.
The minimum inhibitory concentration (MIC) results from these QC strains are then compared against the CLSI® QC VITEK® 2 Results provided in the VITEK® 2 card's package insert, ensuring the accuracy and reliability of the test system. Accordingly, we have clarified this point and sources within the manuscript, specifically in lines 181-185.
Comment 2: Line 206: Please add P-value to statistical analysis.
Response 2: Agree. We have added the P-value to the statistical analysis. We have clarified this point within the manuscript, specifically in line 236.
Comment 3: Please ensure consistency in the use of the bacterial name throughout the manuscript. Use either Escherichia coli or E. coli consistently, but not both interchangeably within the text. The same applies to other bacterial species.
Response 3: Agree. We have revised the manuscript to ensure consistent use of bacterial names throughout, addressing this point.
Comment 4: Line 129: Please provide the date and purpose of the study in the ethical approval section. Also, please add the consent information.
Response 4: Thank you for your valuable comment. We have clarified this point within the manuscript, specifically in lines 118 - 124.
This research was a retrospective study, approved by the Kasetsart University Institutional Animal Care and Use Committee (IACUC) under protocol number U1-00453-2558 on September 18, 2024. Our purpose was to investigate the bacterial species causing feline urinary tract infections and their antimicrobial patterns. As such, this study utilized pre-existing, leftover diagnostic specimens (isolated bacterial cultures) and anonymized outpatient (OPD) data from client-owned, non-experimental animals. Our research team did not directly collect samples; instead, sample acquisition was performed solely at the discretion of the attending veterinarian, strictly adhering to established professional standards of care. All samples and data were part of routine, standard veterinary care. No identifiable personal information was collected or used, and all data were handled in strict accordance with ethical guidelines and institutional regulations. Consequently, written informed consent from animal owners was not required for this particular study.

Reviewer 2 Report
Comments and Suggestions for Authors
This manuscript presents a comprehensive, well-organized study investigating bacterial species and their antimicrobial resistance (AMR) patterns in feline urinary tract infections (UTIs) in Thailand. It includes a substantial number of samples (n = 543). It provides valuable data for local empirical therapy recommendations and broader global AMR surveillance. While the study primarily employs conventional microbiological and antimicrobial susceptibility testing methods, the work is nonetheless scientifically valuable and deserves publication. Its relevance lies in the scarcity of region-specific data on antimicrobial resistance in feline urinary tract infections and its direct implications for empirical treatment practices and antimicrobial stewardship. The simplicity of the techniques used does not detract from the importance of the findings, which are highly informative for both veterinary clinicians and public health stakeholders.
The manuscript investigates the prevalence of bacterial species and their AMR profiles in feline urinary tract infections in Thailand, with a particular focus on the implications for empirical antimicrobial therapy.
The manuscript contributes significantly to regional AMR surveillance in companion animals, particularly in Southeast Asia where such data is scarce.
The species-level identification of uncommon pathogens like Corynebacterium urealyticum and Lactococcus garvieae adds novelty. The methodology is standard but sound. Bacterial isolation via cystocentesis and identification through 16S rRNA sequencing is appropriate. The authors might consider clarifying how intrinsic resistance was excluded in their MDR classification and whether confirmatory ESBL testing was performed for E. coli or Klebsiella isolates. However, these are minor suggestions and do not affect the overall validity of the results.
However, comparisons with previous local studies could be better integrated to highlight trends or shifts in pathogen prevalence.
The use of cystocentesis minimizes contamination, and the inclusion of both Gram-positive and Gram-negative bacteria is comprehensive. The method of classifying MDR is standard, but intrinsic resistance exclusions should be more clearly detailed. ESBL identification is only briefly discussed, consider including confirmation protocols if available. Recurrent UTI data is underexplored. Consider discussing risk factors or underlying causes that might contribute to reinfection patterns.
Figures 1–6 are informative but would benefit from:
Consolidation or complementary summary visuals. More descriptive figure legends for standalone clarity.
Conclusions are consistent with the data and respond directly to the study’s objectives. The authors appropriately highlight the limited effectiveness of some first-line antimicrobials and the need to update empirical treatment protocols in light of local resistance patterns.
Author Response
Response to Reviewer 2's comments
Comments: This manuscript presents a comprehensive, well-organized study investigating bacterial species and their antimicrobial resistance (AMR) patterns in feline urinary tract infections (UTIs) in Thailand. It includes a substantial number of samples (n = 543). It provides valuable data for local empirical therapy recommendations and broader global AMR surveillance. While the study primarily employs conventional microbiological and antimicrobial susceptibility testing methods, the work is nonetheless scientifically valuable and deserves publication. Its relevance lies in the scarcity of region-specific data on antimicrobial resistance in feline urinary tract infections and its direct implications for empirical treatment practices and antimicrobial stewardship. The simplicity of the techniques used does not detract from the importance of the findings, which are highly informative for both veterinary clinicians and public health stakeholders. The manuscript investigates the prevalence of bacterial species and their AMR profiles in feline urinary tract infections in Thailand, with a particular focus on the implications for empirical antimicrobial therapy.
The manuscript contributes significantly to regional AMR surveillance in companion animals, particularly in Southeast Asia where such data is scarce. The species-level identification of uncommon pathogens like Corynebacterium urealyticum and Lactococcus garvieae adds novelty.
The methodology is standard but sound. Bacterial isolation via cystocentesis and identification through 16S rRNA sequencing is appropriate. The authors might consider clarifying how intrinsic resistance was excluded in their MDR classification and whether confirmatory ESBL testing was performed for E. coli or Klebsiella isolates. However, these are minor suggestions and do not affect the overall validity of the results. However, comparisons with previous local studies could be better integrated to highlight trends or shifts in pathogen prevalence. The use of cystocentesis minimizes contamination, and the inclusion of both Gram-positive and Gram-negative bacteria is comprehensive. The method of classifying MDR is standard, but intrinsic resistance exclusions should be more clearly detailed. ESBL identification is only briefly discussed; consider including confirmation protocols if available. Recurrent UTI data is underexplored. Consider discussing risk factors or underlying causes that might contribute to reinfection patterns. Figures 1–6 are informative but would benefit from: Consolidation or complementary summary visuals. More descriptive figure legends for standalone clarity. Conclusions are consistent with the data and respond directly to the study’s objectives. The authors appropriately highlight the limited effectiveness of some first-line antimicrobials and the need to update empirical treatment protocols in light of local resistance patterns
Response: We sincerely thank the reviewer for their comprehensive and insightful evaluation of our manuscript. We appreciate their positive assessment of our study's scientific value, particularly its contribution to local empirical therapy recommendations and broader antimicrobial resistance (AMR) surveillance in feline urinary tract infections in Thailand. Their recognition of the scarcity of region-specific data and the novelty of identifying uncommon pathogens is especially encouraging. We have carefully considered all comments and suggestions, which have helped us to improve the manuscript.
Comments 1: Clarifying MDR Classification and ESBL Testing (Reviewer Comment: “The authors might consider clarifying how intrinsic resistance was excluded in their MDR classification and whether confirmatory ESBL testing was performed for E. coli or Klebsiella isolates. The method of classifying MDR is standard, but intrinsic resistance exclusions should be more clearly detailed. ESBL identification is only briefly discussed; consider including confirmation protocols if available.”)
Response 1: Thank you for your valuable feedback regarding our MDR classification and ESBL testing. We agree that these points require clearer elaboration.
Regarding MDR Classification: We have ensured that our multidrug-resistant (MDR) definition explicitly considers only acquired antimicrobial resistance, excluding intrinsic resistance. This is achieved in two ways:
The VITEK® 2 antimicrobial susceptibility testing system, as detailed in Materials and Methods, lines 198-205, does not report susceptibility results for organisms with known intrinsic resistance to certain drugs.
We have revised lines 223-224 to explicitly state that our MDR definition specifically excludes intrinsic resistance, which refers to a bacterial species' natural, inherent, and predictable resistance to certain drugs.
Regarding ESBL Testing: For ESBL screening, we relied exclusively on the VITEK® 2 system. We acknowledge that molecular confirmation methods or bla gene detection were not employed in this research. This clarification has been added to lines 216-219 of the manuscript.
Comments 2: Comparisons with previous local studies could be better integrated to highlight trends or shifts in pathogen prevalence.
Response 2: Thank you for this valuable suggestion. We agree that integrating comparisons with previous local studies enhances the discussion. We have specifically clarified contrasting susceptibility, particularly for Escherichia coli to amoxicillin-clavulanic acid, found in the Chiang Mai study. This comparison adds an important dimension to our discussion of empirical treatment choices and is now detailed in the manuscript, lines 564-567.
Comments 3: Consider discussing risk factors or underlying causes that might contribute to reinfection patterns.
Response 3: We sincerely appreciate your valuable suggestion regarding the discussion of potential risk factors contributing to reinfection patterns. We agree this would significantly enrich the understanding of our findings.
However, as this was a retrospective study, we relied exclusively on secondary data. This inherently limited the completeness of patient histories. While chronic kidney disease and urethral stenosis were frequently observed comorbidities among cats diagnosed with UTIs in our dataset, the absence of comprehensive medical histories for all 95 patients unfortunately precluded a thorough analysis of the association between specific underlying conditions and the occurrence of initial or recurrent infections.
Comments 4: Recurrent UTI data is underexplored.
Response 4: Thank you for your valuable feedback. We acknowledge and agree that the recurrent UTI data were underexplored and that discussing associated risk factors would have been beneficial.
We sincerely apologize that our study, being retrospective and based solely on pre-existing, anonymized diagnostic and outpatient (OPD) data, couldn't provide a deeper analysis in these areas. We lacked access to specific clinical outcome information or detailed patient histories, such as the treating antimicrobial given, which are necessary to thoroughly analyze recurrent UTI patterns or associated risk factors.
We have acknowledged this as a limitation in our discussion and suggested it as a crucial area for future prospective research in lines 599-604.
Comments 5: Figures 1–6 are informative but would benefit from: Consolidation or complementary summary visuals. More descriptive figure legends for standalone clarity.
Response 5: Thank you for this excellent suggestion. We agree that our figures would benefit from consolidation and improved legends. We have now combined the visual data into a new figure, Figure 2-4 (lines 298-312), to present the information more efficiently and concisely. Additionally, we've enhanced the descriptive nature of all figure legends, including the heat maps (Figures 5-10, lines 353-401 and 433-467), ensuring they offer standalone clarity and provide a comprehensive understanding of the presented data without requiring extensive reference to the main text. We believe these changes significantly improve the overall presentation and accessibility of our visual findings.

Reviewer 3 Report
Comments and Suggestions for Authors
The research addresses about the key question on prevalence and pattern of antibiotic resistance pathogens in feline urinary tract infections.
The original data fulfils the following scientific gap which is not much explored in Thailand
Key epidemiological pattern of urinary tract infections in cats in Thailand
Prevalent pathogens associated with urinary tract infections in cats in Thailand
Antibiotic sensitivity patterns of the key pathogens associated with urinary tract infections in cats in Thailand
This data adds the new knowledge to the subject area and which will be highly useful for the clinicians to make decisions for treating feline urinary tract infections in Thailand and also helpful for the policy makers to mitigate the AMR
Overall the manuscript is scientifically sound and without any flaws, however I recommend the following minor suggestions
The introduction part is lengthy which could be concised and paragraph regarding choice of treatment of feline urinary tract infections could be merged with discussion
The sampling methodology in the materials and methods section need to be explained
It’s better to add a diagram on methodology
Author Response
Response to Reviewer 3's comments
The research addresses about the key question on prevalence and pattern of antibiotic resistance pathogens in feline urinary tract infections. The original data fulfils the following scientific gap which is not much explored in Thailand Key epidemiological pattern of urinary tract infections in cats in Thailand Prevalent pathogens associated with urinary tract infections in cats in Thailand. Antibiotic sensitivity patterns of the key pathogens associated with urinary tract infections in cats in Thailand. This data adds the new knowledge to the subject area and which will be highly useful for the clinicians to make decisions for treating feline urinary tract infections in Thailand and also helpful for the policy makers to mitigate the AMR. Overall the manuscript is scientifically sound and without any flaws, however I recommend the following minor suggestions. The introduction part is lengthy which could be concised and paragraph regarding choice of treatment of feline urinary tract infections could be merged with discussion. The sampling methodology in the materials and methods section need to be explained It’s better to add a diagram on methodology.
Response: We sincerely thank the reviewer for their thorough review and positive assessment of our manuscript. We are particularly pleased that they recognize our work addresses key gaps in the understanding of feline urinary tract infections and antimicrobial resistance (AMR) patterns in Thailand, providing valuable data for clinicians and policymakers. We appreciate the acknowledgment that our data adds new knowledge to the field and is scientifically sound.
We have carefully considered all minor suggestions to further enhance the manuscript.
Comments 1: Introduction Length and Treatment Discussion Placement (Reviewer Comment: "The introduction part is lengthy which could be concised and paragraph regarding choice of treatment of feline urinary tract infections could be merged with discussion.")
Response 1: Thank you for your valuable feedback regarding the Introduction's length and the suggestion to integrate the discussion on feline urinary tract infection (UTI) treatment choices.
We have now modified the Introduction section to improve readability and sharpen the focus on our study's objectives. Additionally, we've integrated relevant paragraphs concerning the choice of treatment for feline UTIs by reducing their presence in the Introduction (lines 105-110) and merging specific parts into the Discussion section (lines 636-638). This allows for a more integrated analysis of our findings within the context of current treatment strategies.
Comments 2: The sampling methodology in the materials and methods section need to be explained It’s better to add a diagram on methodology
Response 2: Thank you for your valuable feedback. We agree that a clearer explanation of our sampling methodology, supplemented with a visual aid, will significantly enhance the readability and understanding of our Materials and Methods section.
We have added a new diagram (Figure 1; lines 136-151) to visually represent this workflow. This flowchart clearly illustrates the entire process, starting from the standard clinical veterinary diagnosis and subsequent bacterial isolation from urine, and extending to our research phase, where these bacterial isolates are received and further analyzed for our study.
